# FILL IN THE BLANK: EXPLORING AND ENHANCING LLM CAPABILITIES FOR BACKWARD REASONING IN MATH WORD PROBLEMS

## ABSTRACT

While *forward reasoning* (i.e., find the answer given the question) has been explored extensively in the recent literature, backward reasoning is relatively unexplored. We examine the *backward reasoning* capabilities of LLMs on Math Word Problems (MWPs): given a mathematical question and its answer, with some details omitted from the question, can LLMs effectively retrieve the missing information?

In this paper, we formally define the backward reasoning task on math word problems and modify three datasets to evaluate this task: GSM8k, SVAMP and MultiArith. Our findings show a significant drop in the accuracy of models on backward reasoning compared to forward reasoning across four SOTA LLMs (GPT4, GPT3.5, PaLM-2, and LLaMa). Utilizing the specific format of this task, we propose three novel techniques that improve performance: *Rephrase* reformulates the given problem into a forward reasoning problem, *PAL-Tools* combines the idea of Program-Aided LLMs to produce a set of equations that can be solved by an external solver, and *Check your Work* exploits the availability of natural verifier of high accuracy in the forward direction, interleaving solving and verification steps. Finally, realizing that each of our base methods correctly solves a different set of problems, we propose a novel Bayesian formulation for creating an ensemble over these base methods aided by a verifier to further boost the accuracy by a significant margin. Extensive experimentation demonstrates that our techniques successively improve the performance of LLMs on the backward reasoning task, with the final ensemble-based method resulting in a substantial performance gain compared to the raw LLMs with standard prompting techniques such as chain-of-thought.

## 1 INTRODUCTION

Large language models (LLMs) (Brown et al., 2020; OpenAI, 2023; Anil et al., 2023) have shown remarkable versatility, excelling in various tasks like sentence completion, question answering, and summarization. They have been successfully applied to mathematical reasoning, specifically in solving *Math Word Problems* (Kushman et al., 2014; Roy & Roth, 2018), where the goal is to produce the answer given an elementary school-level mathematics question. We refer to this task as *Forward Reasoning*. This problem has received significant attention in the recent literature (Lu et al., 2022), and specific datasets (Cobbe et al., 2021; Roy & Roth, 2015; Patel et al., 2021) have been proposed to serve as benchmarks for this task. Performance of powerful LLMs such as GPT-4 OpenAI (2023) with techniques such as Chain-of-Thought (Wei et al., 2022) and Self-Verification (Zhou et al., 2023) on some of these datasets are more than $90\%$ (Lu et al., 2022).

In this work, we are interested in a slightly different problem, which is the problem of *backward reasoning*: given an MWP, with one of the numerical quantities omitted from the question, and the answer to the original question, what is the value of the omitted numerical quantity? While this problem of backward reasoning has been studied in the literature in the context of improving the performance of forward reasoning (Weng et al., 2022), to the best of our knowledge, there is no existing work that explicitly aims to solve this problem analyzing its hardness and providing solutions thereof. We believe this is an interesting problem because (1) It is a matter of study that even for humans, whether forward reasoning and backward reasoning have different complexities (Ramful & Olive, 2008; Rivera, 2008), and we would like to ask the same question in the context of LLMs (2)

Assuming we establish that backward reasoning is a harder problem, how can we design techniques to improve performance on this task, that specifically exploit the problem structure of backward reasoning and the availability of the forward direction answer? (3) The backward reasoning problem can be seen as a special case of abduction, with a unique answer, and it is interesting to explore this connection, since LLMs have not been explored as much for this important class of abductive reasoning problems (Bhagavatula et al., 2020; Qin et al., 2020; 2022).

In order to establish that backward reasoning is indeed a significantly harder problem than forward reasoning, we create backward reasoning problems from three benchmark datasets (Cobbe et al. (2021); Roy & Roth (2015); Patel et al. (2021)), and test the performance of various LLMs on this task. Table 1 presents the results (see Section 3 for details). We observe a significant drop in the performance of all the LLMs compared to forward reasoning, the drop being as high as 40% in most cases. Having established the difficulty of our proposed task, we propose a set of techniques for improving the performance on this task

Firstly, we propose to rephrase the backward reasoning as a forward reasoning problem since LLMs are better at forward reasoning compared to backward reasoning. This can be done by substituting some unknown quantity $x$ in the blank, and then asking LLM to produce the value of $x$, given the answer. Next, we combine the idea of Program-Aided-Language Models, with an external solver since backward reasoning problems have a specific mathematical structure, which is not amenable to simple forward reasoning. We refer to this method as *PAL-Tools*. Thirdly, we can use the LLM as a verifier to check whether the solution to the backward problem is correct. Verification can be done iteratively until the backward reasoning model produces an answer which is declared correct by the verifier. We refer to this approach as *Check your Work*. Finally, we make use of Bayesian modeling to propose a novel method for ensembling base methods, which exploits the accuracy of each of the base methods computed on a small hold-out set, to compute answer probability. Further, we use a high-accuracy verifier based on forward reasoning and compute the probability of the final answer given the ensemble probability and the confidence of the verifier in the answer.

Our results show the benefit obtained by each of our base methods, with Verify and Reprompting performing best overall. Finally, we obtain a further improvement of up to 10% using our Bayesian model, which ensembles techniques using a verifier. The overall gain in accuracy obtained by our methods touches 45% for some of the datasets, compared to directly running the original backward problem on the LLM. We perform additional analysis on the models to explain our results.

To summarize our contributions: (1) we create three different datasets for backward reasoning, establish the difficulty of the backward reasoning problem, and a task of interest in its own right (2) We propose three different base methods, and one ensemble-based method via the use of a verifier, to improve the performance on the backward task. (3) We perform additional analysis giving further insights into the performance of the proposed models.

## 2 RELATED WORK

A mathematical word problem (MWP) (Lu et al., 2022) consists of a description in natural language that expresses the relation between various entities and quantities, followed by a query for an unknown quantity, as shown in Figure 1. One can answer the question by representing the relationship between the entities and quantities through a set of equations and then solving these equations. Solving MWPs necessitates a semantic understanding of the natural language description. Initial works (Kushman et al., 2014; Koncel-Kedziorski et al., 2015; Roy & Roth, 2018) to solve MWPs involve parsing the natural language description and utilizing statistical learning techniques to identify suitable templates for generating answers. Subsequently, following the triumph of sequence-to-sequence (Seq2Seq) neural models (Sutskever et al., 2014) in machine translation and other NLP tasks, the encoder-decoder framework (Wang et al., 2017; Ling et al., 2017; Li et al., 2020; Shen et al., 2021; Jie et al., 2022) is employed to directly translate the natural language description in MWPs into equations.

Recently, the strongest performance on MWPs has been given by large pre-trained language models like GPT-4 OpenAI (2023), PaLM (Anil et al., 2023), and GPT-3 (Brown et al., 2020). These models leverage the power of few-shot in-context examples and employ prompting methods like CoT (Wei et al., 2022), all without requiring any modifications to their parameters. One class of

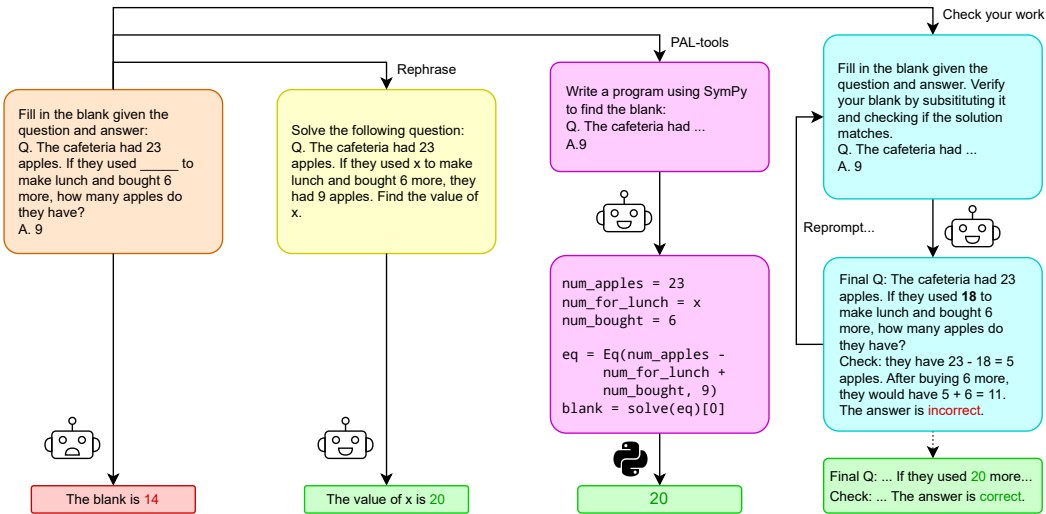

Figure 1: A summary of the prompting techniques we propose

Table 1: Performance of various models on the Math backward reasoning task, compared to their accuracies on the forward reasoning task of solving the original problem. Figures with $^\dagger$ are taken from Zheng et al. (2023)

| Model | GSM8k | | SVAMP | | MultiArith | |
|---|---|---|---|---|---|---|
| | forward | backward | forward | backward | forward | backward |
| GPT-4 | 92.8 | 38.6 | $90.5^\dagger$ | 43.9 | $97.8^\dagger$ | 54.8 |
| GPT-3.5-turbo | 58.4 | 10.8 | 79.1 | 20.4 | 97.0 | 13.8 |
| PaLM-2 | 60.5 | 15.2 | 73.7 | 11.2 | 95.7 | 6.3 |
| LLaMa-2-70B | 37.0 | 6.8 | 70.3 | 20.3 | 89.2 | 11.0 |

techniques (Madaan et al., 2023; Welleck et al., 2023) using LLMs involves verifying the answer provided by the Language Model, either using the model itself or external verifiers such as compilers or proof checkers. If the answer is incorrect, the model is re-prompted, optionally with suggestions on improving its output. This prompting continues until the model generates the correct output. Other techniques, such as Progressive Hint Prompting (Zheng et al., 2023) iteratively pass the model's previous answers to itself as hints. Iterative prompting techniques like (Wang et al., 2023) do not use a verifier; instead, they sample multiple hypotheses from the model and select the answer using majority voting.

Our work can be seen as a special case of abductive reasoning with a unique answer. Abductive reasoning (Bhagavatula et al., 2020; Qin et al., 2020; 2022) involves inferring which of several explanations is the most plausible. Prior work on abductive reasoning has focused mostly on text-based reasoning under constraints. In the context of arithmetic reasoning tasks Weng et al. (2022) has utilized abductive reasoning to enhance forward reasoning accuracy. In contrast, our work addresses backward reasoning as an independent problem. Our primary interest lies in analyzing the inherent complexities of backward reasoning and devising more effective solutions to tackle it.

## 3 BACKWARD REASONING TASK

A forward or the typical Mathematical Word Problem (MWP) consists of a question $Q_f$, which we call a forward question, and its corresponding answer $A_f$. The forward question is a textual representation of the MWP. It is typically composed of one or more sentences and encompasses various elements, including numbers, operations, and textual information, all represented by tokens within the question. A *numeric token* is defined as a token that encodes a numeric quantity, such as

5, 37, or 'half.' These tokens encode the values 5, 37, and $0.5$, respectively. We define a backward MWP as a tuple $(Q, A_f)$, where $Q$ is obtained from the forward question $Q_f$ by replacing one numerical quantity with a blank. The goal of solving the backward MWP is to find out the value of the numerical quantity that was blanked out using backward reasoning. By backward reasoning, we mean the process of using the provided answer $A_f$ and the context provided by the question $Q$ to deduce the missing numerical quantity to arrive at the given answer. Since there is a unique answer for every question, we measure accuracy on this task by the number of questions on which the model is able to provide the correct numeric value of the blank.

Table 1 compares the performance of four state-of-the-art (SOTA) language models on forward and backward reasoning tasks. We observe a significant drop in backward reasoning accuracy compared to forward reasoning accuracy across all models. The experiments were conducted using the chain of thought prompts defined in Wei et al. (2022). The few-shot examples used in the chain of thought prompts were modified for the backward reasoning task following the procedure described above. It is evident from the results that Large Language Models (LLMs) are not as proficient in backward reasoning compared to forward reasoning, indicating the difficulty of the task as defined above.

## 4 PROPOSED APPROACHES

**Rephrasing:** Our first base method to tackle the challenging backward reasoning problem involves a problem transformation through rephrasing. This transformation effectively converts the complex backward reasoning task into a more manageable forward reasoning problem. Consequently, we employ the LLM to solve this transformed forward reasoning problem instead of the original and inherently more difficult backward reasoning challenge.

Given a backward MWP $(Q, A_f)$, we ask the language model to produce a rephrased question $R$, which incorporates the forward answer $A_f$ into the question $Q$ and changes the objective of the question from finding the answer $A_f$ to finding the value of the blank. We then ask the language model to solve the rephrased problem $R$ instead of the original backward problem.

Our experiments in Table 2 show that converting the backward MWP into an algebraic MWP by replacing the blank with $x$ and asking the LLM to find the value of $x$ as shown in Figure 1 gives us better results than converting to a simple MWP which asks us to find the value of the unknown quantity.

**PAL-Tools:** The second base method that we propose combines Program-aided language models (PAL) (Gao et al., 2023) with the techniques of framing equations and solving them using SymPy (Meurer et al., 2016), (He-Yueya et al., 2023) (referred to hereafter as *tools*). We observe that PAL performs very poorly at the backward reasoning task due to the non-sequential nature of the task. When PAL is used for backward reasoning, LLMs must first rearrange the terms of the equation instead of merely finding a formula to use given the variables in the problem. By only using the framing equations, we present the backward reasoning problem to the LLM in a structured mathematical form, making it easier for them to understand and generate code or solutions based on the given equations.

Recently, Schick et al. (2023) have shown that LLMs can utilize external tools and APIs to solve various downstream tasks. Motivated by this, we leverage the LLM's ability to call SymPy functions to overcome their inability to manipulate equations to isolate the unknown variable. Our prompt frames the problem as an equation in $x$ and asks the model to generate code that solves the equation using Sympy's `solve` method, as shown in Figure 1. Parallel work by Zhou et al. (2023), which uses GPT-4's code interpreter to both solve and verify solutions with code, shows similar improvements on the forward task.

**Reprompting and Verification:** Our third base method *Reprompting and Verification* is based on recently proposed framework of SELF-REFINE (Madaan et al., 2023). SELF-REFINE is an iterative prompting technique that cycles between refinement and feedback until a predetermined condition is attained. We modify the SELF-REFINE to perform backward reasoning on MWP. We use PAL-tools as the base model within Self-Refine and refine the programs generated by PAL-tools with feedback from the language model as was done in (Madaan et al., 2023).

Table 2: Improvements in accuracy with various prompting strategies

| Strategy | Shots | GSM8k$_B$ | SVAMP$_B$ | MultiArith$_B$ |
|---|---|---|---|---|
| CoT | 8-Shot | 10.77 | 20.40 | 14.50 |
| PAL | 4-Shot | 9.27 | 20.90 | 18.17 |
| Tools | 3-Shot | 31.45 | 43.50 | 71.83 |
| Rephrase (Linguistic) | 8-Shot | 19.65 | 32.60 | 40.50 |
| Rephrase (Algebraic) | 8-Shot | 36.12 | 37.80 | 71.67 |
| PAL-tools | 2-Shot | 37.26 | 41.00 | 79.67 |
| PAL-tools (Rephrased) | 2-Shot | 37.89 | 42.10 | 77.67 |
| PAL-tools | 4-Shot | 37.11 | 42.70 | 80.50 |
| PAL-tools (Rephrased) | 4-Shot | **48.74** | **51.10** | 84.50 |
| Self-Refine (Rephrased) | 2-Shot | 40.17 | 49.70 | 77.50 |
| Check your Work (Rephrased) | 8-Shot | 41.82 | 47.40 | **84.83** |

Our method works as follows:

1. We start by instructing the LLM to solve the backward MWP $(Q, A_f)$ and provide the numeric value of blank in question $Q$.

2. In this step, we verify the answer generated by the LLM. To do this, we substitute the blank in $Q$ with the LLM's answer, creating a modified question, denoted as $Q'$. We then instruct an LLM to generate the answer $A'$ of this newly formed forward question.

3. We compare the answer $A'$ obtained from $Q'$ with the original answer $A_f$. If $A'$ matches with $A_f$ our method stops.

4. If, upon comparison, the answers $A'$ and $A_f$ do not match, we go back to step 1 and ask LLM to generate alternate hypotheses by adding the feedback in obtained step 2.

Note that the second step above, which involves verifying the answer, can be performed either by the same or a more powerful LLM. We refer to this method as *Check your work*. An illustration of this process is provided in Figure 1.

## 5 Ensembling and Verification

In this section, we present a simple approach for ensembling the set of base methods described previously followed by use of verifier to further improve the accuracy. We assume that we are solving the backward reasoning problem, where given a backward question $Q$, we are interested in finding the correct answer $A$ to fill in the blank in the backward question $Q$.

### 5.1 Computing the Prior Distribution

To create an ensemble, we use the idea of frequency voting. Assume that we are given a set of models $\{M_1, M_2, \cdots, M_k\}$. Given a model $M_i$, we run the model $M_i$ on the question $Q$ $r$ times, to get a multi-set of answers $\{A_{ij}\}_{j=1}^r$.

We compute the prior probability $P(A_l \mid Q)$ as simply the fraction of times $A_l$ appears as the answers in the union of multiset of answers produced by each model for question $Q$ : $P(A_l = A \mid Q) = \sum_{i,j} \mathbb{1}[A_{ij} = A]/kr$. This generalizes the idea of majority voting to compute a prior distribution over all the answers produced by a set of models in a given set of runs. This ensemble can be used for (1) Simple Voting: Output the answer with the highest probability (2) Sampling: Output an answer based on its prior probability (3) Prior: Use this as a prior distribution, to compute another posterior with additional knowledge as described below.

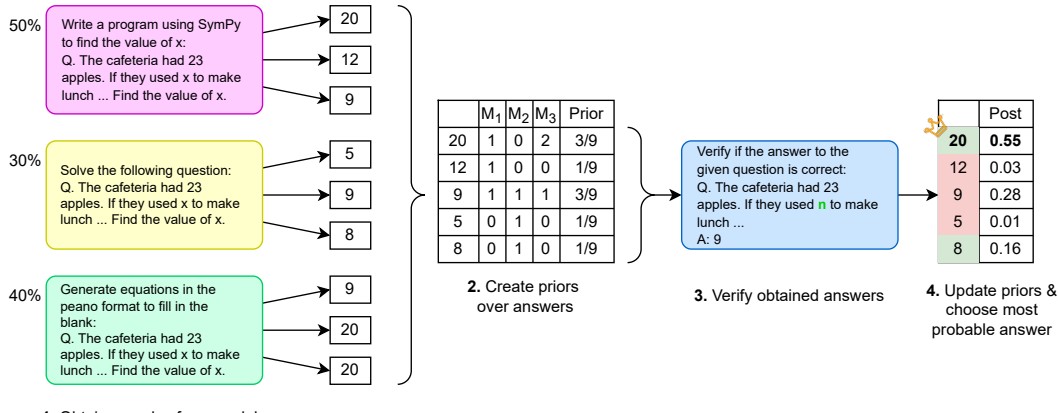

Figure 2: An illustrative example of how the ensembling of base models works together with a verifier.

## 5.2 IMPROVEMENTS USING A VERIFIER

Next, let us assume we have a verifier $V$, such that given a question $Q$ and an answer $A_l$, $V$ gives a Boolean output $Z_l$. This output is equal to 1 if $A_l$ is the correct answer to the question according to the verifier, and 0 otherwise. Let $P_1(Z_l|A_l = A, Q)$ denote the distribution over $V$'s outputs when $A_l = A$. Similarly, let $P(Z_l|A_l \neq A, Q)$ denote the distribution over $V$'s outputs when $A_l \neq A$. We estimate these distributions by computing the accuracy of the verifier on the holdout set $S'$, and supplying a set of answers produced by the $k$ models, each run $r$ times on each $Q \in S'$, along with the gold answer $A$. Please note that we could also use the probability $P_1(Z_l|A_l = A, Q)$ provided by the verifier, but this probability may not be well calibrated (Jiang et al., 2021; Zhao et al., 2021), i.e., the model's probability estimates may not accurately reflect the true likelihood of the answer being correct. Therefore, we opted to use the holdout set to estimate these probabilities.

The probability that $A_l$ is the correct answer for a question $Q$, given the prior probability and the verifier output $Z_l$, by application of Bayes Rule, can be written as

$$P(A_l = A \mid Z_l, Q) = \frac{P(Z_l \mid A_l = A, Q) \, P(A_l = A \mid Q)}{P(Z_l \mid A_l = A, Q) P(A_l = A \mid Q) + P(Z_l \mid A_l \neq A, Q) P(A_l \neq A \mid Q)} \quad (1)$$

The first term in the numerator captures the prior probability over the verifier output given the correct answer and is computed based on the holdout set as described earlier. The second term is simply the prior probability as computed in section 5.1. The denominator is simply the normalization constant: the first term is the same as in the numerator. In the second term, the first sub-term captures the prior probability of verifier output given an incorrect answer and is obtained from the validation set. The second sub-term can be computed simply by subtracting the prior distribution over answers. Figure 2 illustrates the process of how ensembling works using an example.

## 6 EXPERIMENTS

### 6.1 SETUP

In our experimental setup, we work with three primary datasets: GSM8k (Cobbe et al., 2021), MultiArith (Roy & Roth, 2015), and SVAMP (Patel et al., 2021). We transform the examples in these datasets into backward tasks, resulting in the creation of three new datasets: GSM8k$_B$, SVAMP$_B$, and MultiArith$_B$. We have experimented with four SOTA LLMs: GPT-4, GPT-3.5-Turbo (OpenAI, 2023), PaLM-2 (Anil et al., 2023) and LLaMa-2 (Touvron et al., 2023). The first three models are utilized via their official model APIs, while for LLaMa-2, we use the 70-billion-parameter model quantized to 4-bit using GPTQ (Frantar et al., 2022). Apart from Table 1, all other tables use GPT-3.5-Turbo as the base model, unless mentioned otherwise.

Table 3: Results of Ensembling

|  | GSM8k$_B^\dagger$ | SVAMP$_B^\dagger$ | MultiArith$_B^\dagger$ |
|---|---|---|---|
| CoT | 35.67 | 37.78 | 69.60 |
| Tools | 41.81 | 48.11 | 72.00 |
| PAL-Tools | 48.55 | 45.00 | 81.50 |
| Ensemble | **65.33** | **66.67** | **92.60** |

We work with 4 different prompting techniques: Chain of Thought (Wei et al., 2022), PAL (Gao et al., 2023), Tools (He-Yueya et al., 2023) and SELF-REFINE (Madaan et al., 2023). Prompts and in-context examples for the prompting techniques are taken from their original works. The in-context examples are modified for the backward setting as discussed in Section 4. Examples of the prompts used are given in Appendix A.2.

## 6.2 APPROACHES

We test our approaches using GPT-3.5-Turbo as the base model. The results are highlighted in Table 2. We classify the prompts into four categories:

1. **Baseline prompts**: Prompts whose in-context examples were adapted to the backward task without changing the technique.

2. **Rephrased prompts**: Prompts that solve the rephrased question using chain-of-thought reasoning. Used in proposed base method rephrasing.

3. **PAL-Tools prompts**: Prompts that use PAL-tools and their variants.

4. **Adaptive prompts**: Prompts used for resampling another hypothesis from the LLM if the verification condition is not satisfied in the base method Reprompting and Verification.

For rephrasing, we find that models perform better when the rephrased problem has the blank replaced with $x$ compared to the baseline prompt This is because the relationship between the missing value and the equations that models need to frame in order to solve the forward problem is explicit. Also, for the baseline prompt, the model requires inferring the relationship between the forward answer and the equations they need to frame to obtain it, which may introduce ambiguity and reduce accuracy.

For programmatic techniques, we see that PAL-Tools is more performant than Pal or Tools taken separately. We hypothesize that LLMs encounter a larger amount of code in their training data compared to the Peano solutions that He-Yueya et al. (2023) use in their prompt, and this implies that they may find it easier to frame equations in a programming language rather than in a domain-specific language specified via few-shot examples. We find that 4-shot PAL-tools with rephrasing performs the best on GSM8k$_B$ and SVAMP$_B$, while Check your Work prompting marginally outperforms PAL-tools on MultiArith$_B$. We also see that more in-context examples strongly assist in rephrasing, as 4-shot rephrased PAL-Tools strongly outperforms 2-shot rephrased PAL-Tools when compared to the base version without any rephrasing.

Finally, our proposed Check your Work method does marginally better than with our designed prompts than prompts given the original Self-Refine paper (Madaan et al., 2023), with the added benefit of much lower cost. While Self-Refine's feedback prompts are quite large, containing extensive feedback, check your work only appends the final question and chain of thought for verifying the final question to the few-shot examples. This reduces cost as well as inference time. We also see that Self-Refine outperforms the comparable PAL-tools variant, indicating that feedback from the language model does assist in correcting reasoning.

## 6.3 ENSEMBLING

For our ensemble experiments, we selected the 100 examples from the datasets as holdout sets to compute the prior probabilities of the models and the verifier's accuracy. The models we included in the ensemble are rephrased versions of three of our strongest single-prompt models: Chain-of-Thought, Tools, and PAL-Tools. We then evaluated the ensemble on the non-holdout set, which is

Table 4: Improvements using Rephrasing

| Model | Shots | Rephrased | GSM8k$_B$ | SVAMP$_B$ | MultiArith$_B$ |
|---|---|---|---|---|---|
| CoT | 8-shot | No | 10.77 | 20.4 | 14.50 |
| | | Yes | **36.12** | **37.8** | **71.67** |
| | | | (↑25.35) | (↑17.4) | (↑57.17) |
| PAL | 4-shot | No | 9.27 | 20.9 | 18.17 |
| | | Yes | **21.38** | **37.0** | **55.50** |
| | | | (↑12.11) | (↑16.1) | (↑37.33) |
| Tools | 3-shot | No | 31.45 | 43.5 | 71.83 |
| | | Yes | **41.43** | **48.5** | **73.00** |
| | | | (↑9.98 ) | (↑5.0 ) | (↑1.17 ) |
| PAL-tools | 4-shot | No | 37.11 | 42.7 | 80.50 |
| | | Yes | **48.74** | **51.1** | **84.50** |
| | | | (↑11.63) | (↑8.4 ) | (↑4 ) |

| actual | model | |
|---|---|---|
| | positive | negative |
| positive | 75.94 | 24.05 |
| negative | 7.39 | 92.61 |

Table 5: Confusion matrix for verifying problems and their solutions on GSM8k$_B$, normalized across rows

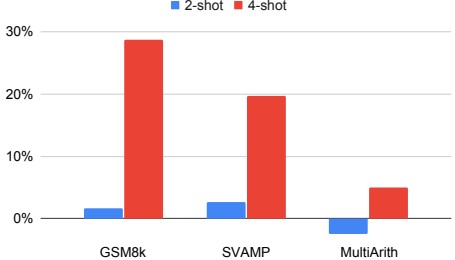

Figure 3: Relative performance increase rephrasing brings to PAL-tools when 4 shots are used compared to 2.

denoted with a [†] symbol in Table 3 to show the results. We observed that the accuracy on backward MWP via ensembling surpasses the forward accuracy of Chain-of-Thought by up to $6\%$.

# 7 ANALYSIS

**How much does rephrasing help?** Since rephrasing is a strategy that can be applied across multiple techniques, we analyze the extent of accuracy gains obtained via rephrasing by applying it independently to four techniques: CoT, PAL, Tools, and PAL-Tools. The results are shown in Table 4. Rephrasing improves the accuracy of every technique that it is applied to. We see larger gains with rephrasing in weaker methods, such as CoT. We also see that rephrasing has higher gains in datasets where the problems are harder, such as in GSM8k compared to SVAMP.

**Is verifying easier than solving?** In the third step of the ensembling method, we try to verify whether the blank provided is correct by solving the resulting forward problem after substituting the blank. There are two settings in which we can verify this: 1) We ask the model to solve this new question, and compare whether the answer obtained is same as original answer $a$. 2) We give the original answer $a$ to the model and ask it to check whether it is the answer obtained for new question.

To find which method is better at correctly verifying the blank, we check the accuracy of GPT-3.5-turbo on GSM8k in setting 2. In the first pass, we provide the correct blank and in the second pass, we provide an incorrect blank formed by multiplying $z \in \{2 \ldots 10\}$ with the correct blank. The confusion matrix obtained is shown in Figure 5. It is observed that the accuracy of setting 2 is higher than the forward reasoning accuracy of GPT-3.5-turbo. Hence we use that as the verification method for ensembling.

Carla just gave birth to identical octuplets. She dresses 3/4 of them in purple and _____ in blue. If all the blue wearers and 1/3 of the purple wearers also wear bows, what is the percentage chance a baby wearing a bow is wearing purple?

Ian has a board that is 40 feet long. He decides to make a cut so he can have _____ pieces. The longer piece is 4 times longer than the shorter piece. How long is the longer piece?

Table 7: Examples of questions in GSM8k$_B$ that don't require the answer to find the value of the blank

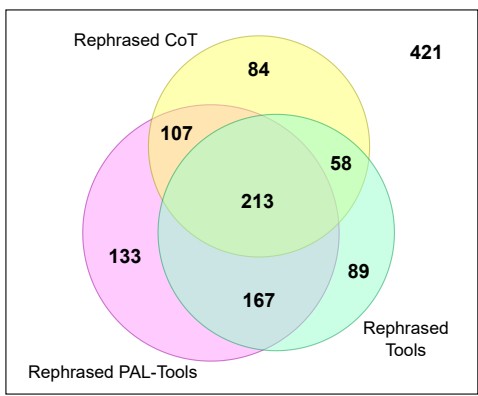

Figure 4: Venn diagram of the subsets of problems of GSM8k$_B$ different prompts are able to solve

**Does the verifier assist in ensembling?** We compare the accuracies obtained by using majority voting with and without the verifier. We find that using a verifier improves the accuracy on GSM8k$_B^\dagger$ and SVAMP$_B^\dagger$ by 7% and on MultiArith$_B^\dagger$ by 0.6%. Since the verifier has a higher accuracy than any of the models we consider, its inclusion inevitably increases the accuracy of any set of methods we choose. Even if we use a noisy verifier, updating our priors based on its results using Bayes' rule ensures that the priors are not changed significantly.

**Do some prompts subsume others?** Let prompts $M_1, M_2, M_3$ be able to solve problems $\mathcal{D}_1, \mathcal{D}_2, \mathcal{D}_3$ respectively, where $\mathcal{D}_i \subseteq D$. If we choose to ensemble these prompts together, then $|\bigcup_j \mathcal{D}_j| > |\mathcal{D}_i|$ for the ensemble to do better. Figure 4 gives an overview of the subsets of GSM8k three prompts, namely Rephrased Chain of Thought, Rephrased PAL-Tools and Rephrased Tools can solve in a single try. We see that even though there is significant overlap between the prompts, the probability of any one of them giving the right answer is 66.9 %, provided we sample from each prompt once. The venn diagram also shows that the subsets of problems different prompts cover is quite disjoint in nature. No prompt can solve all the problems that another prompt can solve.

**Can every blank's answer be determined?** There may be cases where the blank does not directly contribute to the answer, or is irrelevant. In such a case, inferring the value of the blank is not possible given the answer. Even though Cobbe et al. (2021) claim that less than two percent of problems have breaking errors, We sample 50 random examples from GSM8k$_B$that our strongest model solves incorrectly and find that no such problems in the sample we analyse, leading us to believe that the probability of such problems existing in our dataset is little to none.

**Do all blanks need answers to be solved?** In the 50 examples we analyse above, there are 10 examples where the value of the blank can be obtained simply from reading the question, as the question makes implicit assumptions or provides further information that can be used to fill in the blank. Two examples are presented in figure 3. It is surprising that even our strongest model is unable to find the answer to such questions, either as a consequence of its poor reasoning abilities or because we make the dependency between requiring the answer to fill in the blank explicit.

## 8 CONCLUSION

We show that backward reasoning is still a hard problem for large language models, by showing that their accuracy drops significantly on this task compared to forward reasoning. To augment this, we propose multiple methods that can improve their accuracy in this setting and are extensible across preexisting models. We also propose a Bayesian ensembling technique that can combine multiple models in the presence of a noisy verifier. These techniques combined bring backward performance on par with forward reasoning performance. Finally, we analyze the fallacies and pitfalls of each of these techniques and show areas for future improvements.

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

## A APPENDIX

### A.1 DATASET

We consider three datasets of interest: GSM8K (Cobbe et al., 2021), MultiArith (Roy & Roth, 2015) and SVAMP (Patel et al., 2021). All these datasets consist of grade-school arithmetic word problems along with their answers.

#### A.1.1 GENERATION METHODOLOGY

Given a source forward dataset

$$D = \{(Q_i, A_i)_{i=1}^n \mid Q_i \in \Sigma^*, A_i \in \mathbb{R}\}$$

we present a method to create a backward dataset

$$D'_k = \{(Q'_i, A_i, (B_i^0, \ldots, B_i^k))_{i=1}^n \mid Q'_i \in \Sigma^*, A_i, B_i^j \in \mathbb{R}\}$$

To convert $Q_i$ (Source question) to $Q'_i$ (blanked out question) and extract blanks $B_i^0 \ldots B_i^k$, we split $Q_i$ into it's constitutent tokens based on a delimiter, usually space. We then consider all numeric tokens, which are defined as tokens that encode a number. Numeric tokens may be alphanumeric, such as \$42, 80% or 3.14, or they may be alphabetic, such as three, twice or half. Using this heuristic for numeric tokens, we ignore the first numeric token and extract the next $k$ tokens sequentially. If we are unable to extract $k$ tokens, then we skip that question and answer pair. It is worth noting that for the datasets we use, $k = 1$, that is we only consider the problem of backwardly inferring one missing number in the question, given the answer. Solving the $n > 1$ case would require first checking if a unique solution exists and is a topic for future work.

| Dataset | $N$ | Example |
|---|---|---|
| GSM8k | 1272[*] | Kylar went to the store to buy glasses for his new apartment. One glass costs \$5, but every second glass costs only 60% of the price. Kylar wants to buy 16 glasses. How much does he need to pay for them? |
| GSM8k$_B$ | 1272 | Q : Kylar went to the store to buy glasses for his new apartment. One glass costs \$5, but every second glass costs only _____% of the price. Kylar wants to buy 16 glasses. How much does he need to pay for them? 
 A : 64 |
| SVAMP | 1000 | 28 children were riding on the bus. At the bus stop 82 children got on the bus while some got off the bus. Then there were 30 children altogether on the bus. How many more children got on the bus than those that got off? |
| SVAMP$_B$ | 1000 | Q : 28 children were riding on the bus. At the bus stop _____ children got on the bus while some got off the bus. Then there were 30 children altogether on the bus. How many more children got on the bus than those that got off?" 
 A : 2 |
| MultiArith | 600 | Lana picked 36 tulips and 37 roses to make flower bouquets. If she only used 70 of the flowers though, how many extra flowers did Lana pick? |
| MultiArith$_B$ | 600 | Q : Lana picked 36 tulips and _____ roses to make flower bouquets. If she only used 70 of the flowers though, how many extra flowers did Lana pick? 
 A : 3 |

Table 7: Sample questions from the datasets we consider

The reason we choose to blank out only numeric tokens rather than an entire phrase or sentence is to make the task of validation easier. An alternative that was explored was phrase masking. However, phrase masking would lead to generations that would not be verifiable with perfect accuracy, and multiple possible generations for each question. The benefit of number masking is that quantities can be compared to each other without loss of accuracy, and every question-answer pair has a unique blank.

### A.1.2    GENERATION RESULTS

Using the above method, we were able to convert 1272 of the 1319 question and answer pairs in GSM8k to backward reasoning problems, and all 1000 and 600 pairs in SVAMP and MultiArith respectively.

### A.1.3    DATASET EXAMPLES

Some examples of the datasets under consideration are shown in Table 7. Note that for GSM8k, the original dataset contains 1319 sample problems but our dataset generation method for the backward task filters out 47 of them. For comparability with the backward task, we have used the 1272 common examples of this dataset. We use all the problems in SVAMP and MultiArith.

### A.2    PROMPTS

We construct prompts by changing the original examples of the papers we consider. We show one to two in-context examples of each prompt. The remaining examples may be seen in our code.

```
1  Rephrase the given blanked question and answer pairs and then find the
   solution to the rephrased question. Give your answer as either a number
   or a decimal (no fractions). Follow the format specified in the examples
   below:
2
3  Q: There are 15 trees in the grove. Grove workers will plant ______ trees
   in the grove today. After they are done, how many trees would be there?
4  A: 21
5  Rephrased: There are 15 trees in the grove. Grove workers will plant x
   trees in the grove today. After they are done, there would be 21 trees.
   Find the value of x.
6  Answer: There are 15 trees originally, Then there were 21 trees after
   some more were planted. So there must have been x = 21 - 15 = 6 trees.
   The answer is 6.
7
8  Q: If there are 3 cars in the parking lot and ______ more cars arrive,
   how many cars are in the parking lot?
9  A: 5
10 Rephrased: If there are 3 cars in the parking lot and x more cars
   arrive, there are 5 cars in the parking lot. Find the value of x.
11 Answer: There are originally 3 cars. x more cars arrive. 3 + x = 5, so x
   = 5 - 3 = 2. The answer is 2.
12
13 ...
14
15 Q: {{question}}
16 A: {{answer}}
17 Rephrased:
```

Figure 5: Rephrasing with x

```
1 You are given a math question with a blank value and an answer. Solve it
   step b step to find the value of blank. Strictly follow the format given
   in the examples below.
2
3 Question: Ben has four boxes with ten basketball cards in each box. Ben
   received ______ cards from his mother. If he gives 58 cards to his
   classmates, how many cards does he has left?
4 Answer: 22
5
6 Peano solution:
7
8
9 Let a be number of boxes [[var a]]. We have [[eq a = 4]].
10 Let b be number of cards in each box [[var b]]. We have [[eq b = 10]].
11 Let c be number of cards Ben initially has [[var c]]. We have [[eq c = a
   * b]].
12 Let d be cards received from mother [[var d]].
13 Let e be cards given to classmates [[var e]]. We have [[eq e = 58]].
14 Let f be cards left [[var f]]. From given Answer, we have [[eq f = 22]].
15 We have [[eq d = f + e - c]]
16 The answer is the value of d [[answer d]].
17
18
19
20
21 Question: Natalia sold ______ clips to her friends in April, and then she
   sold half as many clips in May. How many clips did Natalia sell
   altogether in April and May?
22 Answer: 72
23
24 Peano solution:
25
26
27 Let a be number of clips Natalia sold in April [[var a]].
28 So number of clips Natalia sold in May are half of a.
29 Let b be number of clips sold altogether [[var b]]. From given Answer,
   we have [[eq b = 72]].
30 We have [[eq a = b / (1 + 1/2)]]
31 The answer is the value of a [[answer a]].
32
33 ...
34
35 Q: {{question}}
36 A: {{answer}}
37
38 Peano solution:
```

Figure 6: Tools

```
1   Rephrase the given blanked question and answer pairs and then solve it
    step b step to find the value of blank. Strictly follow the format given
    in the examples below.
2
3   Question: Ben has four boxes with ten basketball cards in each box. Ben
    received ______ cards from his mother. If he gives 58 cards to his
    classmates, how many cards does he has left?
4   Answer: 22
5
6   Rephrased: Ben has four boxes with ten basketball cards in each box. Ben
    received x cards from his mother. If he gives 58 cards to his
    classmates, he has 22 cards left. Find the value of x.
7   Peano solution:
8
9
10  Let a be number of boxes [[var a]]. We have [[eq a = 4]].
11  Let b be number of cards in each box [[var b]]. We have [[eq b = 10]].
12  Let c be number of cards Ben initially has [[var c]]. We have [[eq c = a
    * b]].
13  Let x be cards received from mother [[var x]].
14  Let d be total cards with Ben [[var d]]. We have [[eq d = c + x]]
15  Let e be cards given to classmates [[var e]]. We have [[eq e = 58]].
16  Let f be cards left [[var f]]. From given Answer, we have [[eq f = 22]].
17  As cards left are also total cards - cards given to classmates, we have
    [[eq f = d - e]]
18  The answer will be the value of x [[answer x]].
19
20
21
22
23
24  Question: Natalia sold ______ clips to her friends in April, and then she
    sold half as many clips in May. How many clips did Natalia sell
    altogether in April and May?
25  Answer: 72
26
27  Rephrased: Natalia sold clips to x of her friends in April, and then she
    sold half as many clips in May. Find the value of x such that she sold a
    total of 72 clips altogether in April and May.
28  Peano solution:
29
30
31  Let x be number of clips Natalia sold in April [[var x]]
32  Let a be number of clips Natalia sold in May [[var a]]. We have [[eq a =
    x / 2]].
33  Let b be number of clips sold altogether [[var b]]. From given Answer,
    we have [[eq b = 72]].
34  As clips sold altogether are also the sum of clips sold in April and
    May, we have [[eq b = x + a ]]
35  The answer will be the value of x [[answer x]].
36
37  ...
38
39  Q: {{question}}
40  A: {{answer}}
41
42  Rephrased:
```

Figure 7: Tools with Rephrasing

```
1  Rephrase the given blanked question and answer pairs and then find the
   solution to the rephrased question. Write a python function that finds
   the value of x by solving step by step. Make sure you name your method
   finding_x. A few examples are given below.:
2
3  Question: Ben has four boxes with ten basketball cards in each box. Ben
   received ______ cards from his mother. If he gives 58 cards to his
   classmates, how many cards does he has left?
4  Answer: 22
5  Rephrased: Ben has four boxes with ten basketball cards in each box. Ben
   received x cards from his mother. He gives 58 cards to his classmates.
   He has 22 cards left.
6  Program:
7  ```python
8  def finding_x():
9      num_boxes = 4
10     cards_per_box = 10
11     # cards_received_from_mother = x – This line is commented because x
   is unknown
12     # hence the variable cards_received_from_mother can't be used in
   R.H.S. of any calculation
13     cards_given_to_classmates = 58
14     cards_left = 22
15     cards_in_boxes = num_boxes * cards_per_box
16     total_cards_before_given_to_classmates = cards_given_to_classmates +
   cards_left
17
18     cards_received_from_mother = total_cards_before_given_to_classmates
   – cards_in_boxes
19     return cards_received_from_mother
20 ```
21
22 Question: Olivia has $23. She bought ______ bagels for $3 each. How much
   money does she have left?
23 Answer: 8
24 Rephrased: Olivia has $23. She bought x bagels for $3 each. She has $8
   left. Find the value of x.
25 Program:
26 ```python
27 def finding_x():
28     money_initial = 23
29     # num_of_bagels = x – This line is commented because x is unknown
30     # hence the variable num_of_bagels can't be used in R.H.S. of any
   calculation
31     bagel_cost = 3
32     money_left = 8
33     money_spent = money_initial – money_left
34
35     num_of_bagels = money_spent / bagel_cost
36     return num_of_bagels
37 ```
38
39 Question: {{question}}
40 Answer: {{answer}}
41 Rephrased:
```

Figure 8: PAL with Rephrasing

```
1  You are given a math question with a blank value and an answer. Write a
   python function called solution() using sympy that assumes the value of
   the blank is x and creates an equation in x that is solved by
   sympy.solve. Return the value of the blank. You may assume the
   neccessary libraries are imported. Strictly follow the format given in
   the examples below, as the method will be executed with the same name.
2
3  Q: Ben has four boxes with ten basketball cards in each box. Ben
   received _____ cards from his mother. If he gives 58 cards to his
   classmates, how many cards does he has left?
4  A: 22
5  Program:
6  ```python
7  def solution():
8      num_boxes = 4
9      cards_per_box = 10
10     total_cards_in_boxes = num_boxes * cards_per_box
11     cards_from_mother = x
12     cards_given_to_classmates = 58
13     cards_left = 22
14
15     equation = Eq(cards_from_mother + total_cards_in_boxes,
   cards_given_to_classmates + cards_left)
16     blank = solve(equation)[0]
17
18     return blank
19 ```
20
21 Q: Natalia sold clips to _____ of her friends in April, and then she
   sold half as many clips in May. How many clips did Natalia sell
   altogether in April and May?
22 A: 72
23 Program:
24 ```python
25 def solution():
26     april_clips = x
27     may_clips = april_clips / 2
28     total_clips = 72
29
30     equation = Eq(april_clips + may_clips, total_clips)
31     blank = solve(equation)[0]
32
33     return blank
34 ```
35 ...
36
37 Q: {{question}}
38 A: {{answer}}
39 Program:
```

Figure 9: PAL-Tools

```
1  Rephrase the given blanked question and answer pairs and then write a
   python function called solution() to find the value of x in the
   rephrased question. Return the value of x. You may assume the neccessary
   libraries are imported. Strictly follow the format given in the examples
   below, as the method will be executed with the same name.
2
3  Q: Ben has four boxes with ten basketball cards in each box. Ben
   received ______ cards from his mother. If he gives 58 cards to his
   classmates, how many cards does he has left?
4  A: 22
5  Rephrased: Ben has four boxes with ten basketball cards in each box. Ben
   received x cards from his mother. He gives 58 cards to his classmates.
   He has 22 cards left. Find the value of x.
6  Program:
7  ```python
8  def solution():
9      num_boxes = 4
10     cards_per_box = 10
11     total_cards_in_boxes = num_boxes * cards_per_box
12     cards_from_mother = x
13     cards_given_to_classmates = 58
14     cards_left = 22
15
16     equation = Eq(cards_from_mother + total_cards_in_boxes,
   cards_given_to_classmates + cards_left)
17     blank = solve(equation)[0]
18
19     return blank
20 ```
21
22 Q: Natalia sold ______ clips to  her friends in April, and then she sold
   half as many clips in May. How many clips did Natalia sell altogether in
   April and May?
23 A: 72
24 Rephrased: Natalia sold x clips to her friends in April, and then she
   sold half as many clips in May. Natalia sells 72 clips altogether in
   April and May. Find the value of x.
25 Program:
26 ```python
27 def solution():
28     april_clips = x
29     may_clips = april_clips / 2
30     total_clips = 72
31
32     equation = Eq(april_clips + may_clips, total_clips)
33     blank = solve(equation)[0]
34
35     return blank
36 ```
37
38 ...
39
40 Q: {{question}}
41 A: {{answer}}
42 Rephrased:
```

Figure 10: PAL-Tools with Rephrasing

```
1  Fill in the blank given the question and answer examples below. Give
   your answer as either a number or a decimal (no fractions). Check your
   work by substituting your answer in the blank, solving the question and
   comparing to the original answer. Follow the format specified in the
   examples below:
2
3  Q: There are 15 trees in the grove. Grove workers will plant ______ trees
   in the grove today. After they are done, how many trees would be there?
4  A: 21
5  Answer: There are 15 trees originally, Then there were 21 trees after
   some more were planted. So there must have been 21 - 15 = 6 trees. The
   blank is 6.
6  Final question: There are 15 trees in the grove. Grove workers will
   plant 6 trees in the grove today. After they are done, how many trees
   would be there?
7  Check: There would be 15 + 6 = 21 trees in total. The original answer
   was 21. This matches the original answer.
8
9  Q: If there are 3 cars in the parking lot and ______ more cars arrive,
   how many cars are in the parking lot?
10 A: 5
11 Answer: There are originally 3 cars. There are finally 5 cars, so 5 - 3
   = 2 cars arrived. The blank is 2.
12 Final question: If there are 3 cars in the parking lot and 2 more cars
   arrive, how many cars are in the parking lot?
13 Check: There would be 3 + 2 = 5 cars in the parking lot. The original
   answer was 5. This matches the original answer.
14
15 ...
16
17 Q: {{question}}
18 A: {{answer}}
19 Answer:
```

Figure 11: Check your work

```
1  Fill in the blank given the question and answer examples below. Give
   your answer as either a number or a decimal (no fractions). Check your
   work by substituting your answer in the blank, solving the question and
   comparing to the original answer. Follow the format specified in the
   examples below:
2
3  Q: There are 15 trees in the grove. Grove workers will plant ______ trees
   in the grove today. After they are done, how many trees would be there?
4  A: 21
5  Rephrased: There are 15 trees in the grove. Grove workers will plant x
   trees in the grove today. After they are done, there would be 21 trees.
   Find the value of x.
6  Answer: There are 15 trees originally, Then there were 21 trees after
   some more were planted. So there must have been x = 21 – 15 = 6 trees.
   The answer is 6.
7  Final question: There are 15 trees in the grove. Grove workers will
   plant 6 trees in the grove today. After they are done, how many trees
   would be there?
8  Check: There would be 15 + 6 = 21 trees in total. This matches the
   original answer.
9
10 Q: If there are 3 cars in the parking lot and ______ more cars arrive,
   how many cars are in the parking lot?
11 A: 5
12 Rephrased: If there are 3 cars in the parking lot and x more cars
   arrive, there are 5 cars in the parking lot. Find the value of x.
13 Answer: There are originally 3 cars. x more cars arrive. 3 + x = 5, so x
   = 5 – 3 = 2. The answer is 2.
14 Final question: If there are 3 cars in the parking lot and 2 more cars
   arrive, how many cars are in the parking lot?
15 Check: There would be 3 + 2 = 5 cars in the parking lot. This matches
   the original answer.
16
17 ...
18
19 Q: {{question}}
20 A: {{answer}}
21 Rephrased:
```

Figure 12: Check your work with Rephrasing

```
You are given a math question with a blank value and an answer. Rephrase
the given blanked question and answer pairs and then write a python
function called solution() to find the value of x in the rephrased
question. Return the value of x. You may assume the neccessary libraries
are imported. Strictly follow the format given in the examples below, as
the method will be executed with the same name.

Q: Ben has four boxes with ten basketball cards in each box. Ben
received _____ cards from his mother. If he gives 58 cards to his
classmates, how many cards does he has left?
A: 22
Rephrased: Ben has four boxes with ten basketball cards in each box. Ben
received x cards from his mother. He gives 58 cards to his classmates.
He has 22 cards left. Find the value of x.
Program:
```python
def solution():
    num_boxes = 4
    cards_per_box = 10
    total_cards_in_boxes = num_boxes * cards_per_box
    cards_from_mother = x
    cards_given_to_classmates = 58
    cards_left = 22

    equation = Eq(cards_from_mother + total_cards_in_boxes,
cards_given_to_classmates + cards_left)
    blank = solve(equation)[0]

    return blank
```

Q: Natalia sold _____ clips to  her friends in April, and then she sold
half as many clips in May. How many clips did Natalia sell altogether in
April and May?
A: 72
Rephrased: Natalia sold x clips to her friends in April, and then she
sold half as many clips in May. Natalia sells 72 clips altogether in
April and May. Find the value of x.
Program:
```python
def solution():
    april_clips = x
    may_clips = april_clips / 2
    total_clips = 72

    equation = Eq(april_clips + may_clips, total_clips)
    blank = solve(equation)[0]

    return blank
```

Q: {{question}}
A: {{answer}}
Rephrased:
```

Figure 13: PAL-Tools with Rephrasing and Self-Refine: init prompt

```
1  You are given a question-answer pair with a blank, and a chain of
   thought (CoT) for filling in the blank. Go through the chain of thought
   step by step and point out mistakes, if any. Provide the final corrected
   answer as shown below.
2
3  Q: Kelly is grocery shopping at a supermarket and is making sure she has
   enough in her budget for the items in her cart. Her 5 packs of bacon
   cost $_____ in total and she has 6 packets of chicken which each cost
   twice as much as a pack of bacon. She also has 3 packs of strawberries,
   priced at $4 each, and 7 packs of apples, each priced at half the price
   of a pack of strawberries. If Kelly's budget is $65 then how much money,
   in dollars, does she have left in her budget?
4  A: 5
5  Rephrased: Kelly is grocery shopping at a supermarket and is making sure
   she has enough in her budget for the items in her cart. Her 5 packs of
   bacon cost a total of x dollars, and she has 6 packets of chicken, each
   costing twice as much as a pack of bacon. She also has 3 packs of
   strawberries priced at $4 each, and 7 packs of apples, each priced at
   half the price of a pack of strawberries. If Kelly's budget is $65, then
   how much money, in dollars, does she have left in her budget?
6
7  Program:
8
9  ```python
10 def solution():
11     bacon_cost = x
12     chicken_cost = 2 * bacon_cost
13     strawberry_cost = 4
14     apple_cost = strawberry_cost / 2
15     budget = 65
16     remaining_budget = budget - (5 * bacon_cost + 6 * chicken_cost + 3 *
   strawberry_cost + 7 * apple_cost)
17
18     equation = Eq(remaining_budget, 5)
19     blank = solve(equation)[0]
20
21     return blank
22 ```
23
24 Correction:
25
26 The rephrased problem does not end with 'Find the value of x'. Instead,
   it asks us to find the money Kelly has left in her budget. This is
   incorrect.
27
28 Let's check the program:
29
30     bacon_cost = x
31     chicken_cost = 2 * bacon_cost
32
33 Looks good
34
35     strawberry_cost = 4
36     apple_cost = strawberry_cost / 2
37
38 Looks good
39
40     budget = 65
41     remaining_budget = budget - (5 * bacon_cost + 6 * chicken_cost + 3 *
   strawberry_cost + 7 * apple_cost)
42
43 The equation formed is correct, as are the number of packs of bacon,
   chicken, strawberry and apples
44
45     equation = Eq(remaining_budget, 5)
46     blank = solve(equation)[0]
47     return blank
48                                    23
49 The equation has been equated to the correct answer, and the program
   returns the blank.
50
```

```
1  Final Rephrased Problem:
2  Kelly is grocery shopping at a supermarket and is making sure she has
   enough in her budget for the items in her cart. Her 5 packs of bacon
   cost a total of x dollars, and she has 6 packets of chicken, each
   costing twice as much as a pack of bacon. She also has 3 packs of
   strawberries priced at $4 each, and 7 packs of apples, each priced at
   half the price of a pack of strawberries. If Kelly's budget is $65, then
   she has $5 left in her budget after shopping. Find the value of x.
3
4  Final Program:
5  ```python
6  def solution():
7      bacon_cost = x
8      chicken_cost = 2 * bacon_cost
9      strawberry_cost = 4
10     apple_cost = strawberry_cost / 2
11     budget = 65
12     remaining_budget = budget - (5 * bacon_cost + 6 * chicken_cost + 3 *
   strawberry_cost + 7 * apple_cost)
13
14     equation = Eq(remaining_budget, 5)
15     blank = solve(equation)[0]
16
17     return blank
18  ```
19  ...
20
21  Q: {{question}}
22  A: {{answer}}
23  Rephrased: {{rephrased}}
24  Program:
25  ```python
26  {{program}}
27  ```
28
29  Correction:
```

Figure 15: PAL-Tools with Rephrasing and Self-Refine: feedback prompt continued

