# OpenReview forum: "Fill in the Blank: Exploring and Enhancing LLM Capabilities for Backward Reasoning in Math Word Problems"
_ICLR.cc/2024/Conference — Submitted to ICLR 2024_

### Official Review · Reviewer_AYQE · 2023-10-30

**Soundness:** 2 fair
**Presentation:** 2 fair
**Contribution:** 2 fair
**Rating:** 3
**Confidence:** 5

**Summary:**

This paper provides insight into the relatively unexplored area of backward reasoning in Math Word Problems (MWPs). The authors formally define the task of backward reasoning to derive missing information from given answers and incomplete questions.
The authors modify three datasets to evaluate this task. The experiments show that multiple Large Language Models (LLMs) showed a significant accuracy drop in backward reasoning compared to forward reasoning.
The authors propose three basic prompt methods as improvements, namely “Rephrasing”, “PAL-Tools” and “Reprompting and Verification”. The authors further propose one ensemble-based method via the use of a verifier.
Through extensive experimentation, the authors demonstrate that their techniques substantially enhance LLM performance on the backward reasoning task, especially the ensemble-based method further boost the accuracy by a significant margin.

**Strengths:**

1. The paper explores the relatively understudied area of backward reasoning in mathematical word problems (MWPs) and formalizes the task of backward reasoning.
2. The research methodology is rigorous. The authors used a variety of state-of-the-art prompt techniques for reverse reasoning with LLM, testing them to ensure a thorough evaluation.
3. The paper is systematically structured and clearly distinguishes between problem definition, methodology, experimentation and analysis.

**Weaknesses:**

1. The motivation seems to be Ambiguous. The paper's definition of backward reasoning essentially frames it as a fill-in-the-blank task, rather than a genuine backward reasoning or a broader sense of causal reasoning. In reference [1], a similar task is merely a sub-task in backward verification for verifying forward reasoning. The authors have repurposed it as a new backward reasoning task, which seems redundant and lacks research value. The paper's discussion on the practical application scenarios of this task is insufficient, making it challenging to discern its real-world significance. Moreover, the rationale behind comparing the difficulty levels of backward and forward reasoning remains unexplained.
2. Lack of Methodological Novelty. The three primary methods presented are essentially repurposed from forward reasoning techniques that have been previously introduced and widely applied in other works. The authors have essentially transformed backward reasoning into forward reasoning, without designing specific methods tailored to the unique characteristics of backward reasoning, thereby undermining the essence of studying backward reasoning. Specifically:
-----The “Rephrasing” method aligns closely with the “Condition Mask Verification” method from reference [1]. This should have been treated as a baseline rather than a novel approach due to its lack of originality.
-----The “PAL-Tools” method is fundamentally the same as the method introduced in reference [2], with the authors merely employing the SymPy library for solving for ‘x’. This approach lacks innovation.
-----The “Reprompting and Verification” method is essentially an inversion of a method from reference [3] used to verify the correctness of forward reasoning, which seems like a forced novelty.
3. While the paper presents a range of experimental results, the depth of analysis behind these results is lacking. For instance, the paper doesn't delve deep into the differences between various prompting techniques and why certain techniques are more effective in specific scenarios. The comparative analysis mostly revolves around forward reasoning methods, lacking a direct comparison with potential backward reasoning techniques, making it difficult for readers to gauge the true advantages of the proposed methods.
4. The authors' modifications to the datasets GSM8k, SVAMP, and MultiArith are minimal, merely replacing numbers in questions with blanks. However, they claim to have created “new datasets”, which seems to exaggerate their contribution.
5. The paper contains many grammatical errors. One is the missing period at the end of the paragraph “In order to establish … on this task”, and the second is the incorrect punctuation in the paragraph “A forward or the typical … numeric value of the blank” with the misplaced colon in ’half.’.

[1]	Weng, Y., “Large Language Models are Better Reasoners with Self-Verification”, <i>arXiv e-prints</i>, 2022. doi:10.48550/arXiv.2212.09561.
[2]	Luyu Gao and Aman Madaan and Shuyan Zhou and Uri Alon and Pengfei Liu and Yiming Yang and Jamie Callan and Graham Neubig (2022)PAL: Program-aided Language Models International Conference on Machine Learning abs/2211.10435
[3]	Aman Madaan and Niket Tandon and Prakhar Gupta and Skyler Hallinan and Luyu Gao and Sarah Wiegreffe and Uri Alon and Nouha Dziri and Shrimai Prabhumoye and Yiming Yang and S. Welleck and Bodhisattwa Prasad Majumder and Shashank Gupta and A. Yazdanbakhsh and Peter Clark (2023)Self-Refine: Iterative Refinement with Self-Feedback arXiv.org abs/2303.17651

**Questions:**

1. What is the research significance and practical implications of the proposed backward reasoning task? Specifically:
---a) Why have you chosen to define backward reasoning as a fill-in-the-blank task and study the capabilities of LLMs on this particular task?
---b) What motivated the exploration of the relative difficulty between backward and forward reasoning?
---c) Could you elaborate on potential applications of this task in sectors like education, industry, or other domains?
2. I found it challenging to understand the precise mechanism of the “Ensembling” method. Can you offer a more intuitive explanation, especially in the context of Figure 2 in your paper? Moreover:
---a) How did you arrive at the decision to use a holdout set of 100 examples from the datasets?
---b)What criteria or methods were used to select these examples for the holdout set? Is there a specific scientific basis for this choice?
3. Would it be possible to design ablation studies to elucidate why the “Ensembling” method performs better? Can you shed light on the effectiveness of each module and whether the method seamlessly integrates the backward reasoning capabilities of the three basic methods?

---

> ### Author Response · Authors · 2023-11-23
> **Author Response to Reviewer AYQE**
>
> **Comment 1:** The motivation seems to be Ambiguous. The paper's definition of backward reasoning essentially frames it as a fill-in-the-blank task, rather than a genuine backward reasoning or a broader sense of causal reasoning. In reference [1], a similar task is merely a sub-task in backward verification for verifying forward reasoning. The authors have repurposed it as a new backward reasoning task, which seems redundant and lacks research value. The paper's discussion on the practical application scenarios of this task is insufficient, making it challenging to discern its real-world significance. Moreover, the rationale behind comparing the difficulty levels of backward and forward reasoning remains unexplained.
>
> **Response:** The primary motivation of our work is to study the backward reasoning abilities of LLMs as observed in human problem-solving. With these insights, we can not only differentiate between LLMs and human problem-solving but also leverage this understanding to improve LLMs. Enhancing the backward reasoning capabilities of LLMs can result in the development of more sophisticated and context-aware language models, ultimately advancing their performance and applicability in various domains. For this, we have chosen a fill-in-the-blank task on Math Word Problems (MWPs), a particular type of causal reasoning task. We chose MWPs because LLMs, like GPT-4, demonstrate strong forward reasoning performance, making them natural and adept verifiers for backward reasoning on MWPs. In contrast to reference [1], where the backward reasoning task is used for verifying forward reasoning, our goal is to enhance the performance of LLMs on the backward reasoning task itself.
>
>
> **Comment 2:** Lack of Methodological Novelty.
>
> **Response:** We first presented three methods, Rephrase, PAL-Tools, and Check your Work, for solving the backward reasoning task. However, in references [1], [2], and [3], the goal is to solve the forward reasoning task. Then, we proposed a novel Bayesian ensembling technique to combine these methods. This technique leverages pretrained Language Models (LLMs) as naturally available verifiers without the need for explicit training. Our proposed Bayesian ensembling technique is versatile and can be applied to any backward reasoning task when a strong verifier is readily available.
>
>
> **Comment 3:** While the paper presents a range of experimental results, the depth of analysis behind these results is lacking. For instance, the paper doesn't delve deep into the differences between various prompting techniques and why certain techniques are more effective in specific scenarios. The comparative analysis mostly revolves around forward reasoning methods, lacking a direct comparison with potential backward reasoning techniques, making it difficult for readers to gauge the true advantages of the proposed methods.
>
> **Response:** Thanks for the suggestion. We are working on a comparative analysis of various prompting techniques, we hope to upload some of these findings before the end of the discussion phase. Few preliminary analyses is present in response of comment 7. Apart from that, we observed Check your Work method doing well as compared to self-refine because of having the original question and answer in the context history for solving to find x and also having that solution for forming new question and checking it. This is because of the design of the prompt of doing these in the same iteration.
>
> **Comment 4:** The authors' modifications to the datasets GSM8k, SVAMP, and MultiArith are minimal, merely replacing numbers in questions with blanks. However, they claim to have created “new datasets”, which seems to exaggerate their contribution.
>
> **Response:** Thank you for your comment. While we understand your perspective, we would like to clarify that our intention was not to exaggerate the contribution but rather to highlight the modifications made in the existing datasets to make them amenable for backward reasoning. We acknowledge that the term "new datasets" might have caused confusion, and we will adjust the language in the revised version of the paper to better reflect the nature of the modifications made.
>
> **Comment 5:** The paper contains many grammatical errors. One is the missing period at the end of the paragraph “In order to establish … on this task”, and the second is the incorrect punctuation in the paragraph “A forward or the typical … numeric value of the blank” with the misplaced colon in ’half.’.
>
> **Response:** Thank you for your feedback. We will carefully review and address the grammatical errors in the revised version of the paper.

---

> > ### Author Response · Authors · 2023-11-23
> > **Additional Author Response to Reviewer AYQE**
> >
> > **Comment 6:** a) How did you arrive at the decision to use a holdout set of 100 examples from the datasets? ---b)What criteria or methods were used to select these examples for the holdout set? Is there a specific scientific basis for this choice?
> >
> > **Response:** We chose a holdout set of 100 examples as they gave a good enough estimate for verification accuracy on the entire dataset as the LLMs we are working with have high forward reasoning accuracy.
> > We chose the first 100 examples from each dataset in the holdout. As problems are not arranged in any order, this is equivalent to choosing the examples randomly.
> >
> > Furthermore, we conducted an experiment using the ensembling method on randomly shuffling the dataset - i.e. with new set of 100 examples as holdout and we did not see any variance in the accuracies(%) to the precision of e-14.
> >
> > **Comment 7:** Would it be possible to design ablation studies to elucidate why the “Ensembling” method performs better? Can you shed light on the effectiveness of each module and whether the method seamlessly integrates the backward reasoning capabilities of the three basic methods?
> >
> > **Response:** We evaluated all the three methods on some examples, where one method works and the other two fail. Based on that, following were our findings:
> >
> > **Rephrasing method:** It does well on simpler problems. It is prone to making mistakes when the algebraic simplification and numerical calculation of finding x from the equations isn’t straightforward.
> >
> > **Tools method:** As this method uses a solver to find the final values of x, LLM’s shortcoming in complex numerical calculation is bypassed, hence it is able to solve problems which the rephrasing method can’t. Tools method is prone to making errors like missing adding the equation for setting the value of a variable in the solution from the question.
> >
> > **PAL-Tools method:** As this method too uses python interpreter, the shortcoming in numerical calculations is bypassed, enabling it to solve problems that rephrasing method makes mistake in. And as the value of each variable is assigned in python while defining it, PAL-Tools is also able to solve problems that Tools may make mistakes in.
> >
> > Thus, all three underlying methods are prone to different kinds of mistakes while solving. We observe that each method is likely to repeat the mistake in subsequent re-prompts, but the probability of all three methods making mistakes in a problem simultaneously is very less.
> > In ensembling due to presence of a strong verifier the problem is solved correctly even if one of the methods correctly solves the problem in one of its iterations.

---

### Official Review · Reviewer_xzVN · 2023-10-31

**Soundness:** 3 good
**Presentation:** 3 good
**Contribution:** 3 good
**Rating:** 8
**Confidence:** 3

**Summary:**

This paper presents a the backward reasoning task on math word problem solving. The authors conduct extensive experiments on three datasets. The empirical results show the effectiveness of the proposed method. The paper is well written and the solution is clear.

**Strengths:**

1. The authors conduct extensive experiments on three datasets. The empirical results show the effectiveness of the proposed method.
2. The paper is well written and the solution is clear.

**Weaknesses:**

1. The paper explores the LLM capabilities for backward reasoning, it is not clear what the complexity is compared to the traditional math word problem solving models, such as Graph2Tree, GTS etc.
2. The implementation details are not clearly described. For example, GPU and memory size, codes and datasets.

**Questions:**

1. The paper explores the LLM capabilities for backward reasoning, it is not clear what the complexity is compared to the traditional math word problem solving models, such as Graph2Tree, GTS etc.
2. The implementation details are not clearly described. For example, GPU and memory size, codes and datasets.

---

> ### Author Response · Authors · 2023-11-23
> **Author Response to Reviewer xzVN**
>
> Thank you for your positive feedback; we appreciate your recognition of the extensive experiments, clear solution, and well-written paper.
>
> **Comment 1**: The paper explores the LLM capabilities for backward reasoning, it is not clear what the complexity is compared to the traditional math word problem-solving models, such as Graph2Tree, GTS etc.
>
> **Response**: But based on the reviewer comment, we tried to experiment with some of these models. But unfortunately, Graph2Tree requires data preprocessing to convert the solution equations into a certain format, and scripts for which are not provided by the authors. We have sent an email to the authors in this regard. At the same time, we note that these techniques require fully supervised data, in the form of equations which solve the reasoning problem. Since we do not have such a set of equations for the backward task, we may only be able to evaluate this after training on the forward problem. We do not expect this to do well, but we are working on these experiments (assuming we are able to get the code) and plan to add the results before the end of the discussion phase.
> GTS assumes specific linguistic data (Chinese) and it is not clear how to extend this to a general purpose setting in other languages.
>
> **Comment 2**: The implementation details are not clearly described. For example, GPU and memory size, codes and datasets.
>
> **Response**: For all experiments involving closed-source LLMs, we utilized the respective model APIs (OpenAI, Google Bard). For the experiment (only inference), using the Llama-2-70B model, we used a 4-bit GPTQ quantized version of the model and two 40GB NVIDIA A100 GPUs were used for the inference. Details regarding the dataset and prompts used in the experiments can be found in Appendix A. We will release the code and the dataset upon acceptance of the paper.

---

### Official Review · Reviewer_9odL · 2023-11-04

**Soundness:** 2 fair
**Presentation:** 3 good
**Contribution:** 2 fair
**Rating:** 5
**Confidence:** 3

**Summary:**

This paper proposes a new problem called "backward reasoning", namely in a math word problem, given some conditions and the final answer, it aims to infer a missing condition. The authors propose three strategies, including rephrase, code-aid tool and verification, with a final ensemble stage to solve the problem. The proposed approach is then evaluated on three math world problem benchmarks, and several ablation studies are done to further understand the role of each design in the proposed approach.

**Strengths:**

- Three strategies, including rephrase, tool and verification, are proposed to solve the ``backward reasoning'' problem.

- Extensive experiments have been done to demonstrate the effectiveness of the proposed approach.

- The paper is very easy to follow and organized well.

**Weaknesses:**

-  The motivation for the proposed ``backward reasoning'' problem is not very clear to me. In practice, if we want to know a condition, it is natural to just rephrase the question and make {given conditions, final answer} as conditions and the missing condition as a question. In other words, I do not really see the essential of defining a so-called "backward reasoning".

- I have non-trivial concerns about the novelty and contribution of this paper. The "rephrase" techniques of the three proposed solutions that use x to replace the missing condition are not novel, which is already proposed in the advanced chain-of-thoughts methods [1]. Also, I do not see the uniqueness of the proposed ``PAL-Tools'', it is basically a special case of PAL [2]. Similar novelty concerns to the proposed ensemble stage as well, since the ensemble is definitely not a new technique and in most cases, it can improve the performance.

- The ``backward reasoning'' problem and proposed approach are only evaluated on math benchmarks. However, if we target for uncovering a missing condition, it is essential to do evaluations on other types of reasoning tasks, such as commonsense and symbolic.


[1] Fu et al, complexity-based prompting for multi-step reasoning.
[2] Gao et al. PAL: Program-aided Language Models

**Questions:**

Please check the weakness section, my concerns and questions are pretty much there.

---

> ### Author Response · Authors · 2023-11-23
> **Author Response to Reviewer 9odL**
>
> We thank the reviewer for recognizing our extensive experiments, showcasing the effectiveness of our proposed approach, and the paper's clarity and well-organized structure.
>
> **Comment 1:** The motivation for the proposed “backward reasoning” problem is not very clear to me. In practice, if we want to know a condition, it is natural to just rephrase the question and make {given conditions, final answer} as conditions and the missing condition as a question. In other words, I do not really see the essential of defining a so-called "backward reasoning".
>
> **Response:** We appreciate your feedback on the motivation behind the "backward reasoning" problem. The essential motivation lies in capturing a distinct cognitive process where the task involves deducing missing conditions from a given answer, resembling human problem-solving strategies. Human problem-solving and decision-making often involve backward reasoning, a powerful technique in mathematics for problem-solving and theorem proving. Presently, Large Language Models (LLMs) excel in forward reasoning, especially in solving Math Word Problems by identifying answers to given questions. However, the exploration of LLM capabilities in backward reasoning, involving deducing missing conditions from a known answer, is surprisingly limited. In our work, we formally defined the task of backward reasoning to understand the limitation of current LLMs that can allow us to enhance the capabilities of language models in a manner more aligned with human-like deductive reasoning. For more motivation, please see the Second paragraph of the Section 1 Introduction.
>
> **Comment 2:** I have non-trivial concerns about the novelty and contribution of this paper. The "rephrase" techniques of the three proposed solutions that use x to replace the missing condition are not novel, which is already proposed in the advanced chain-of-thoughts methods [1]. Also, I do not see the uniqueness of the proposed "PAL-Tools", it is basically a special case of PAL [2]. Similar novelty concerns the proposed ensemble stage as well, since the ensemble is definitely not a new technique and in most cases, it can improve the performance. - The utility of the ensemble lies in the fact that when verification is easier than backward reasoning, we can boost performance significantly.
>
> **Response:**
> While we agree that rephrasing is already proposed in [1], we note that they do not evaluate the technique specifically on the backward reasoning problem. We will clarify in the updated version of the paper. We argue that PAL-Tools is a significant enhancement from PAL, and this is evident from the results in Table 2, where PAL does significantly worse on all the datasets. Though our key idea in PAL-Tools is simple, that we can write the code which will then call a math solver, it is critical for good performance on the backward reasoning problem. Lastly, regarding ensemble, as the reviewer has already pointed out, the utility of ensemble is in the fact that we naturally get a verifier of high accuracy due to the forward reasoning problem being easier to solve. Further, we believe our Bayesian formulation and the use of small labeled dataset, to estimate the accuracy of the verifier are also novel contributions of our work.
>
> **Comment 3:** The ``backward reasoning'' problem and proposed approach are only evaluated on math benchmarks. However, if we target for uncovering a missing condition, it is essential to do evaluations on other types of reasoning tasks, such as commonsense and symbolic
>
> **Response:**  This is a fair comment. At the same time, we do believe this is the first study of its kind, where we have shown that LLMs significantly suffer on performance in the backward reasoning problem compared to forward reasoning, in this simple setting.  We have formally defined the task, and also given a few methods to enhance the performance of LLMs, showing significant improvement in accuracy. We are currently experimenting on a more challenging version of the task, where we mask out an entire phrase instead of a single number. The preliminary experiments show similar trends in numbers, where we see that there is a significant drop in performance compared to forward reasoning, and some of our methods seem to help. We plan to upload a fuller set of experimental results before the end of the discussion phase.
>
> We believe there are interesting connections of our backward reasoning with problems such as code in-filling [1], or purely common sense abductive reasoning problems, and exploring this connection is a direction for future work.
>
> **References:**
>
> 1. Fried, Daniel, et al. "InCoder: A Generative Model for Code Infilling and Synthesis." ICLR 2022.

---

### Official Review · Reviewer_4wdv · 2023-11-10

**Soundness:** 2 fair
**Presentation:** 2 fair
**Contribution:** 2 fair
**Rating:** 6
**Confidence:** 4

**Summary:**

This paper studied the task of "backward reasoning" in math word problems, to be specific, they "mask"  the numeric token with an underscore. Then they adopt several prompting methods to solve the problem as a forward reasoning problem. The methods include the proposed one, reprompt and verification / check your work. Finally, they propose a Bayesin ensembling technique to ensemble the prompting results.

**Strengths:**

1. Propose the task of backward reasoning.
1. Propose a re-prompt and verification method, similar to self-refine technique to iteratively finding the correct answer as prediction.
2. Propose the Bayesian ensembling technique, to obtain superior performance in Table 4.

**Weaknesses:**

1. Only the verifier with Bayesian ensembling gives some novelty and some insights to the community. But they did not expand more analysis except Table 5 (it should be Table 5 rather than Figure 5 I guess)
2. The other content seems more engineering efforts than scientific efforts. I think the dataset is also really important, which should be described (with more details) in the main paper rather than appendix.
3. Not enough experiments to compare, for example, to show that the ensembling is better, we probably need to compare with other ensembling techniques.
4. I think the paper should focus more on the task itself, rather than propose something and just put some short results.

**Questions:**

1. What's the difference between the reprompt and verification and check your work, are they the same thing? Why we have two names for the same method?
2. How is the proposed ensemble compared with other vanilla ensembling methods? For example, if I just do majority voting.

---

> ### Author Response · Authors · 2023-11-23
> **Author Response to Reviewer 4wdv**
>
> We thank the reviewer for the review, for recognizing our contributions, and providing comments to further improve our paper.
>
> **Comment 1:** Only the verifier with Bayesian ensembling gives some novelty and some insights to the community. But they did not expand more analysis except Table 5 (it should be Table 5 rather than Figure 5 I guess)
>
> **Comment 2:** The other content seems more engineering efforts than scientific efforts. I think the dataset is also really important, which should be described (with more details) in the main paper rather than in the appendix.
>
> **Response to above comments:** We thank the reviewer for acknowledging the significance and relevance of the dataset proposed in our work. We will move the details of the dataset from the appendix to the main paper.
>
> Sorry for the typo, it is Table 5 instead of Figure 5.
>
> We respectfully beg to differ from the reviewer regarding novelty and insights for the community. Current Large Language Models (LLMs) have achieved remarkable results in forward reasoning, particularly in Math Word Problems, where the task involves finding the answer to a given question. In contrast, human problem-solving and decision-making often involve backward reasoning, a powerful technique in mathematics for problem-solving and theorem proving. Surprisingly, the capabilities of LLMs in backward reasoning remain relatively unexplored. So, other than proposing datasets, our other major contributions include:
>
> 1. Our experiment on Math word problems demonstrates the challenges LLMs face in replicating the intuitive backward reasoning abilities inherent in human problem-solving. Our findings shed light on the divergence between LLMs and human cognitive processes. This understanding opens avenues for further research to bridge this gap and advance the capabilities of language models in mimicking human-like reasoning strategies.
> 2. We believe we are the first to formally define the backward reasoning task in math word problems and demonstrate a significant decrease in the accuracy of state-of-the-art Language Models (LLMs) on backward reasoning compared to forward reasoning.
> 3. We have introduced novel techniques that enhance the performance of LLMs on the backward reasoning task.
>
> **Comment 3:** Not enough experiments to compare, for example, to show that the ensembling is better, we probably need to compare with other ensembling techniques. Comparison with  majority voting.
>
> **Response:** We conducted experiments using majority voting and observed that our proposed ensembling method outperforms majority voting. Below table shows the comparison of ensembling method with other methods including majority voting.
>
> |          		 | GSM8k$_\text{B}$ | SVAMP$_\text{B}$ | MultiArith$_\text{B}$ |
> |-------------------|----------------------|------------------------|-----------------------------|
> | CoT      		 | 35.67       		 | 37.78         		 | 69.60              		 |
> | Tools    		 | 41.81       		 | 48.11         		 | 72.00              		 |
> | PAL-Tools		 | 48.55       		 | 45.00         		 | 81.50              		 |
> | Majority Voting   | 58.28       		 | 59.07         		 | 92.00              		 |
> | Ensemble 		 | **65.33**   		 | **66.67**     		 | **92.60**          		 |
>
> Table: Comparison of Ensembling with other methods
>
>
> **Comment 4:** I think the paper should focus more on the task itself, rather than propose something and just put some short results.
>
> **Response:** In this work, we have defined a new task of backward reasoning, proposed a new dataset to test the performance of LLMs on backward reasoning, and suggested novel techniques to improve the performance of LLMs on backward reasoning. Please let us know if you want us to include anything specific regarding the task.
>
> **Comment 5:** What's the difference between the reprompt and verification and check your work, are they the same thing? Why we have two names for the same method?
>
> **Response:** Thank you for your observation, and we apologize for the confusion. Both reprompt and verification and check your work refer to the same method. We will update the paper and ensure consistency by using a unified term for clarity.

---

### Meta-Review · Area_Chair_2RNk · 2023-12-21

**Metareview:**

This paper aims to study a new task called "backward reasoning" on math word problems, which is, given some conditions and the final answer, to infer a missing condition. It modifies three existing datasets, GSM8k, SVAMP and MultiArith, to evaluate this task, and studies three methods (Rephrase, PAL-Tools, Check your work) as well as a Bayesian formulation for ensembling. In general, the reviewers acknowledged that the proposed problem is under-explored currently, the experiments are comprehensive, and the paper is well written and structured. However, it seems the paper suffers from the following major weaknesses: The paper claims to propose "three novel techniques", but a couple reviewers question the novelty by pointing out their similarities to existing work. From the author response, the authors also admit that at least the technique "Rephrase" shouldn't have been claimed as a novel contribution. I tend to agree with the reviewers and think that the authors should better distinguish their techniques from existing ones and avoid overclaiming. Similarly, the paper also claims to "create" three different datasets while what they did is modifying existing datasets in a relatively straightforward manner. Also, some reviewer pointed out the experimental analysis suffers from a lack of depth. Overall, I believe the paper should be carefully revised to more precisely state their novelty and address other comments.

**Justification For Why Not Higher Score:**

Please see the weaknesses summarized above.

**Justification For Why Not Lower Score:**

N/A

---

### Decision · Program_Chairs · 2024-01-16

Reject